# Recapitulation of HDV infection in a fully permissive hepatoma cell line allows efficient drug evaluation

Florian A. Lempp[1,2], Franziska Schlund[1], Lisa Rieble[1], Lea Nussbaum 🆔 [1], Corinna Link[1], Zhenfeng Zhang[1], Yi Ni[1,2] & Stephan Urban[1,2]

Hepatitis delta virus (HDV) depends on the helper function of hepatitis B virus (HBV), which provides the envelope proteins for progeny virus secretion. Current infection-competent cell culture models do not support assembly and secretion of HDV. By stably transducing HepG2 cells with genes encoding the NTCP-receptor and the HBV envelope proteins we produce a cell line (HepNB2.7) that allows continuous secretion of infectious progeny HDV following primary infection. Evaluation of antiviral drugs shows that the entry inhibitor Myrcludex B (IC$_{50}$: 1.4 nM) and interferon-α (IC$_{50}$: 28 IU/ml, but max. 60–80% inhibition) interfere with primary infection. Lonafarnib inhibits virus secretion (IC$_{50}$: 36 nM) but leads to a substantial intracellular accumulation of large hepatitis delta antigen and replicative intermediates, accompanied by the induction of innate immune responses. This work provides a cell line that supports the complete HDV replication cycle and presents a convenient tool for antiviral drug evaluation.

---

[1] Department of Infectious Diseases, Molecular Virology, University Hospital Heidelberg, Heidelberg 69120, Germany. [2] German Centre for Infection Research (DZIF), partner site Heidelberg, Heidelberg 69120, Germany. Correspondence and requests for materials should be addressed to S.U. (email: stephan.urban@med.uni-heidelberg.de)

Hepatitis delta virus (HDV) is a small negative-sense RNA satellite virus that causes acute and chronic infections of the liver[1]. Chronic hepatitis delta (CHD) constitutes the most severe form of viral hepatitis with an even enhanced risk of progression to cirrhosis and hepatocellular carcinoma when compared with hepatitis B virus (HBV) monoinfection. HDV manifests as a co- or superinfection with HBV, as HDV requires the HBV envelope proteins for the formation of viral particles and subsequent receptor-mediated entry into hepatocytes. An estimated 5% of chronic HBV carriers are co-infected with HDV, leading to about 15–20 million CHD patients worldwide[2].

HDV is an enveloped, single-stranded RNA virus carrying the smallest genome of all described mammalian viruses composed of 1679 nucleotides (genotype 1) encoding only one open-reading frame for the hepatitis delta antigen (HDAg). HDAg is expressed in two forms: the small S-HDAg (24 kDa) serves diverse functions during viral replication (e.g. ribonucleoprotein (RNP) formation)[3]. The large L-HDAg (27 kDa), which consists of S-HDAg and carries a C-terminal extension of 19–20 amino acids, plays a crucial role in RNP envelopment. Assembly is mediated via the farnesylation of L-HDAg at cysteine residue C211 within its elongated C terminus[4,5]. Integrated in the host cell-derived membrane of HD virions are the three HBV surface proteins L-, M-, and S-HBsAg. Similar to HBV, HDV is highly hepatotropic, entering the host cell via the receptor sodium taurocholate co-transporting polypeptide (NTCP). However, in contrast to HBV, HDV can also enter and replicate in non-hepatic cells, if NTCP is ectopically expressed[6,7]. Entry requires the specific interaction with the preS1 domain of viral L-HBsAg[8]. The RNP is transported to the nucleus, where it replicates via a double rolling-circle mechanism; the two HDAg isoforms are expressed and form new RNPs that bud into the endoplasmic reticulum, where the HBV envelope proteins are located. The budding process involves interaction between the prenylated Cys-211 of L-HDAg with a cytosolic loop of the small HBsAg[5]. Consequently, HDV depends only on the HBV envelope proteins and does not require active HBV replication within the same cell. Principally, envelope proteins can be provided by transcription of HBV cccDNA in a co-infected cell or they can be expressed from HBV genomic integrates[9,10]. If a hepatocyte has never been infected by HBV, HDV can still enter the cell, replicate in the nucleus, and express HDAg, but a progeny virus can only be formed after a subsequent HBV superinfection[11].

Recently, it was shown that HDV replication itself elicits an innate immune response[12,13], which is triggered by MDA5[14]. Although safe and efficient HBV vaccines also protect against HDV, there is no approved treatment available for CHD. Unfortunately, clinically approved HBV RT inhibitors do not interfere with HBsAg secretion and therefore have no effect on HDV. Interferon-α (IFN-α) is the only drug currently being used for treatment; however, viral response rates are limited and not sustained[15]. In recent years, two novel potent HDV-specific antiviral drugs emerged and are now approaching registration trials: (1) Lonafarnib is an orally active inhibitor of the farnesyl transferase, a cellular enzyme involved in post-translational modification of proteins. It inhibits prenylation of the large hepatitis delta antigen (L-HDAg) and thereby blocks assembly and secretion of virions in the cell[16,17]; (2) Myrcludex B (MyrB) is a peptidic inhibitor of the cellular receptor NTCP. By blocking NTCP, MyrB inhibits viral cell entry and spread of both HBV and HDV via the extracellular route within the liver[18,19].

The development of selective novel antiviral therapies for HDV and the evaluation and mode-of-action studies of currently emerging compounds are presently impaired by the lack of a stable cell culture model that supports the full HDV replication cycle. After the identification of NTCP, several cell lines have been developed that overexpress the receptor, thereby allowing HDV entry, replication, and HDAg expression[8,20,21]. However, due to the lack of encoded HBV envelope proteins, these cell lines do not support HDV progeny assembly and secretion, and can therefore not be used to identify compounds that target these late steps of the viral life cycle. Accordingly, although studied intensively, an $IC_{50}$ of the assembly inhibitor Lonafarnib in an authentic infection or the effect of IFN on HD virion production has never been determined so far.

In this study, we develop a hepatoma cell line supporting the complete HD viral replication cycle by stably co-expressing NTCP and the HBV envelope proteins. We characterize this cell line with regard to NTCP receptor functionality and HBsAg secretion. We use it to study the mode of action and determine the $IC_{50}$ of the three investigational drugs Myrcludex B, Lonafarnib, and IFN-α and discover that inhibition of L-HDAg prenylation leads to a substantial intracellular accumulation of L-HDAg and viral genomes, resulting in an increase of innate immune responses.

## Results

**Establishment of HepNB2.7 cells.** To establish a cell line expressing both NTCP and the HBV envelope proteins, we transduced HepG2-NTCP cells[21] with lentiviruses encoding a subgenomic HBV fragment comprising the L-, M-, and S-HBsAg, as well as the HBx open-reading frames (HB2.7 HBV subgenomic fragment[22,23], Fig. 1a). Since the promoter of the plasmid had been removed, expression of the three HBV envelope proteins was exclusively driven by the authentic HBV promoters/enhancers after integration. After transduction, a cell clone was selected, characterized, and named HepNB2.7. By comparison of HepNB2.7 cells with a corresponding cell line deficient in HBx expression, we could exclude a contribution of HBx to HDV assembly and HBsAg expression.

**HepNB2.7 cells co-express NTCP and the HBV envelope proteins.** As shown by western blot, HepNB2.7 cells expressed comparable amounts of NTCP as the parental HepG2-NTCP cells (Fig. 1b). NTCP was properly folded and localized at the plasma membrane, since it exhibited its natural transporter function, as shown by taurocholate (TC) uptake assay (Fig. 1c, Supplementary Fig. 1). The uptake of the TC substrate could be specifically blocked by the HBV preS1-derived lipopeptide Myrcludex B, indicating a specific ligand–receptor interaction. Although the same cells co-express the NTCP receptor and its ligand L-HBsAg, we observed no interference with NTCP localization and function. Moreover, HepNB2.7 cells, in contrast to HepG2 and HepG2-NTCP cells, expressed and secreted HBsAg (subviral particles), as shown by ELISA of the cell culture supernatant (Fig. 1d). The levels of HBsAg were similar to those observed in HepG2.2.15 or HepAD38 cells that also express HBsAg from endogenous promotors but do not express NTCP[24]. Western blot analysis of the cell lysate (Fig. 1e) indicated that all three forms (L, M, and S) of HBsAg were expressed and properly glycosylated.

**HepNB2.7 cells secrete infectious progeny virus after HDV infection.** To test if the cells are capable of secreting progeny HD virions after initial infection, we inoculated HepG2-NTCP or HepNB2.7 cells with an HDV stock, collected the cell culture supernatant at days 12–14 post infection, and used this supernatant for a secondary infection of HuH7-NTCP cells (Fig. 2a). In order to quantify the infectivity of released virions, we used receptor-expressing HuH7-NTCP cells (secondary infection), as this cell line has been shown highly susceptible for HDV[21]. Cells of the primary infection were fixed at day 14 post infection and

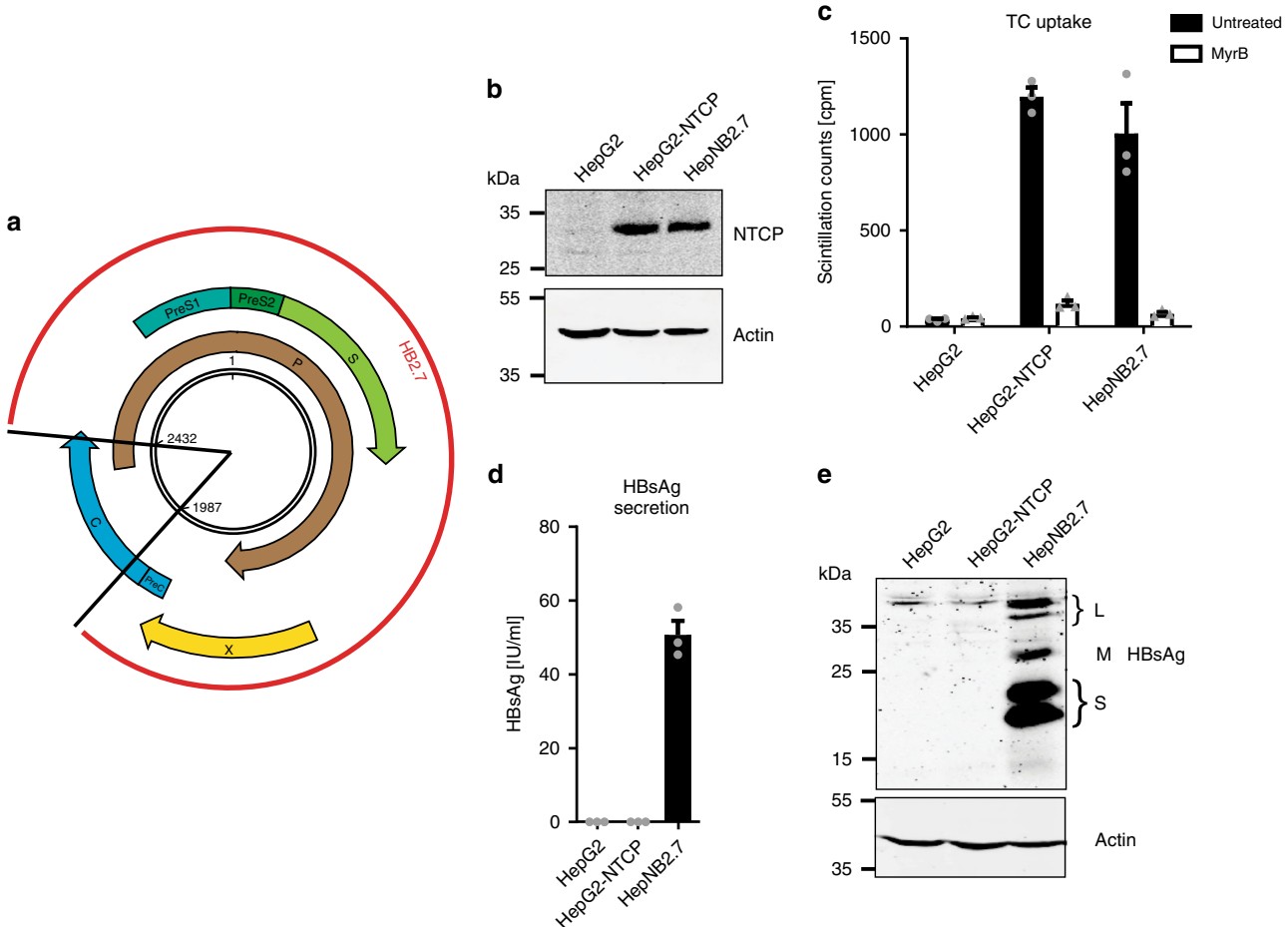

**Fig. 1** Establishment and characterization of the HepNB2.7 cell line. **a** Schematic representation of the HBV genome with its ORFs (arrows) and the HB2.7 subgenomic construct (red) comprising the L-/M-/S-HBsAg and HBx ORFs. **b** NTCP-specific western blot of deglycosylated total cell lysates of parental HepG2, HepG2-NTCP, and HepNB2.7 cells. **c** Uptake of $^3$H-taurocholate in the parental or NTCP-transduced cells (black bars). Uptake was inhibited by pre- and co-incubation with 2 μM Myrcludex B (white bars). **d** ELISA-based quantification of HBsAg in the supernatant of the three cell lines. **e** HBsAg-specific western blot of the total cell lysates

immunostained for HDAg (Fig. 2b, c). Both cell lines were susceptible to HDV and showed comparable infection rates. As intended, HepNB2.7 but not HepG2-NTCP cells secreted infectious progeny virus, demonstrated by secondary infection of HuH7-NTCP cells. In these cells, HDAg was detected, when supernatants from HepNB2.7 but not from HepG2-NTCP cells were used. HBV infection of HepNB2.7 cells was severely impaired compared with HepG2-NTCP with more than 100-fold reduction in released HBeAg upon infection (Fig. 2d). This might be due to a superinfection–exclusion mediated by the L-HBsAg, preventing HBV but not HDV infection (Yi Ni, International HBV Meeting, Oxford, 2012).

**Virus amplification and long-term progeny virus secretion**. To examine the kinetics of progeny virus release after initial infection, HepNB2.7 cells were inoculated with 0.2, 0.6, or 2 IU/cell of HDV and cell culture supernatants were collected every third day for 33 days. The supernatants were analyzed for HBsAg by ELISA (Fig. 3a), secreted viral genomes by RT-qPCR (Fig. 3b), and release of infectious virions as determined by a secondary infection experiment (Fig. 3c). Secreted HBsAg levels increased over time and reached a plateau value of about 200 IU/ml at day 20 post plating. This increase was independent of the amount of viral inoculum, indicating that HDV replication in these cells does not affect HBsAg expression and secretion. In contrast, the amount of

released viral genomes and the released infectivity depended on the viral inoculum, with the highest inoculum (2 IU/cell) resulting in faster secretion kinetics and higher levels reached until day 33 post infection. Secreted infectivity plateaued after 15 days of infection for the 0.6 and 2 IU/cell inocula, while it remained slowly increasing for the 0.2 IU/cell inoculum. In general, we found little differences in viral genome secretion between the 0.6 and 2 IU/cell inocula, indicating that a maximum infection level had been reached. In line with previous studies, the maximum infection level was only 15–30% of total cells, which could not be increased, even when applying higher virus inocula[7].

As HepNB2.7 cells are susceptible to HDV and secrete progeny virus, we expected extracellular viral spread by de novo entry of virions in the system. To test for this, we inoculated cells with a low- or a high-titer HDV inoculum and treated them with the NTCP-specific entry inhibitor Myrcludex B to block NTCP and prevent spread. After 22 days, cells were fixed and HDAg-positive cells were quantified by automated image analysis (Supplementary Fig. 2). Unexpectedly, we found only a 1.2-fold increase in HDAg-positive cells in the untreated versus the MyrB-treated cells, indicating that viral spread via released virions and de novo entry is limited in this system. Generally, susceptibility to HDV infection decreased the longer the cells were held in culture (Supplementary Fig. 3). In order to test whether the lack of PEG during the possible spreading phase limits virus spread, we

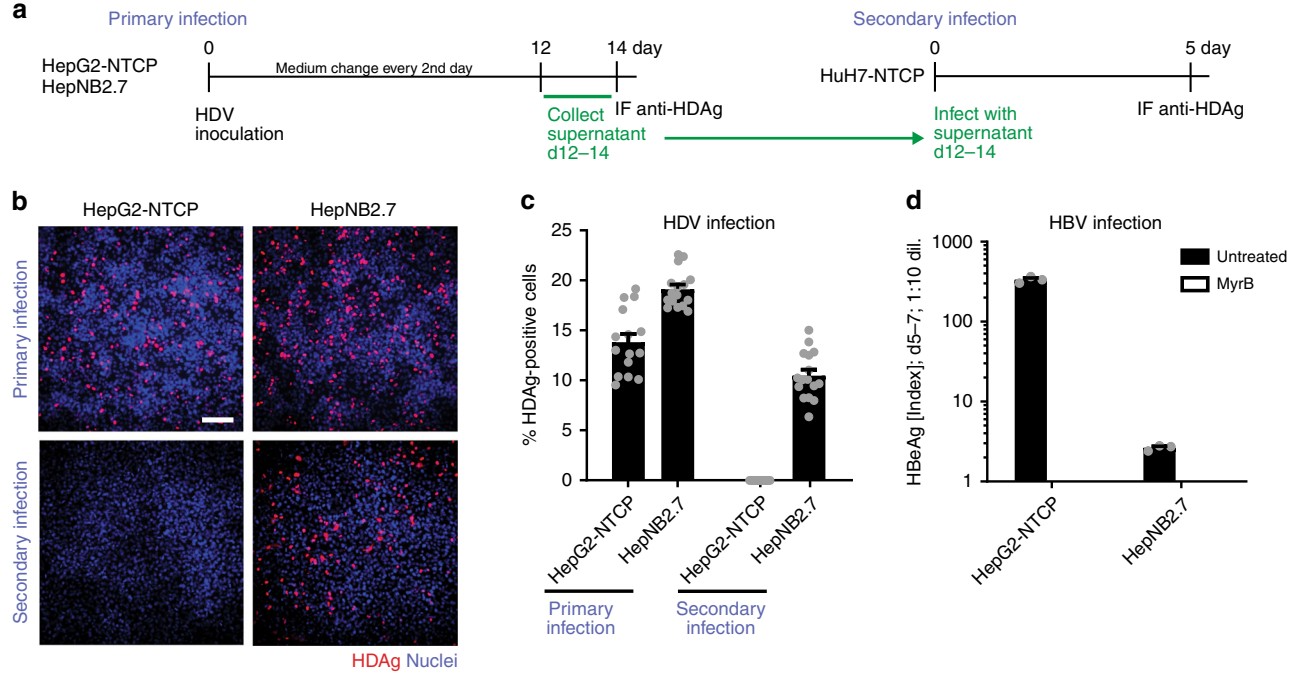

**Fig. 2** HepNB2.7 cells secrete infectious progeny virus after HDV infection. **a** Schematic representation of the experimental layout: HepG2-NTCP or HepNB2.7 cells were inoculated with 2 IU/cell HDV, the supernatant from day 12 to 14 post infection was collected, and one-tenth of the supernatant was used for secondary infection of HuH7-NTCP cells. **b** Cells of primary infection (top) and secondary infection (bottom) were fixed and HDAg (red) was immunostained (scale bar: 100 μm). **c** For infection quantification, 16 images per condition of the immunostained cells were automatically acquired and HDAg-positive cells and nuclei were counted using ImageJ software. **d** HepNB2.7 cells were inoculated with HBV at 150 ge/cell in the presence or absence of 500 nM MyrB. HBeAg in the cell supernatant from day 5 to 7 post infection was quantified by ELISA

cultured the cells in a medium containing 4% PEG in the presence and absence of Myrcludex B. As depicted in Supplementary Fig. 4, continuous addition of PEG after low-level initial infection did not result in enhanced spread in the absence of Myrcludex B.

**Fast quantification of HDV-infected cells by an in-cell ELISA assay.** For antiviral drug screening and compound evaluation studies, middle- to high-throughput assays for the quantification of viral infection are essential. We and others[25] have previously used HDAg-specific immunofluorescence and microscopy-based image readout systems for this purpose. These assays are time consuming, not fully automated, and require elaborate instrumentation. We therefore developed an alternative fast, reliable, and quantitative assay to measure HDAg in infected cells using an in-cell ELISA assay principle[26,27]. The assay combines antibody-mediated labeling of intracellular HDAg in fixed cells with a chemiluminescent readout. For assay evaluation, HepNB2.7 cells seeded in 96-well plates were inoculated with decreasing viral titers starting at 1.25 IU/cell. At day 8 post infection, cells were fixed and analyzed for HDAg by either standard immuno-fluorescence using microscopy-based quantification or by the newly developed in-cell ELISA (Fig. 4). The absolute number of HDAg-positive cells decreased with lower virus inocula (Fig. 4b) which was paralleled by a decrease in the in-cell ELISA RLU signal, measuring the total HDAg per well (Fig. 4a). The determined signal-to-background ratio of 110-fold in the 1.25 IU/cell setting allowed robust quantification with a significant difference to the background up to 0.0012 IU/cell, a setting in which only 1–3 HDAg-positive cells per view field were detectable. Evaluation of the assay in HuH7-NTCP cells, which were used for secondary infection experiments throughout the study, revealed a similar signal-to-background ratio of 94-fold (Supplementary

Fig. 5). Consequently, this in-cell ELISA allowed reliable and rapid quantification of HDV infection in a 96-well plate format and was subsequently applied for $IC_{50}$ determinations of HDV-specific antiviral drugs.

**Mode of action and $IC_{50}$ determination of HDV-specific drugs.** Developmental drugs interfering with late steps of the HDV replication cycle, e.g. Lonafarnib, have never been studied so far in an infection system. As HepNB2.7 cells secrete progeny virus upon primary HDV infection, their effect on de novo virus assembly and secretion was studied. To that aim, HepNB2.7 cells were pre-treated with increasing concentrations of Myrcludex B, Lonafarnib, or IFN-α, subsequently inoculated with HDV in the presence of the respective compounds, and the medium plus fresh compound was changed every second or third day (Fig. 5a). Twelve days post inoculation, the primary infection was analyzed by in-cell ELISA (Fig. 5b). In parallel, cell culture supernatants were used for secondary infection of HuH7-NTCP cells. HDAg was quantified by in-cell ELISA 7 days post infection (Fig. 5c). Cytotoxicity of the drugs was analyzed by WST-1 assay (Fig. 5d).

MyrB completely blocked primary infection with an $IC_{50}$ of 1.2 nM and consequently progeny virus secretion and secondary infection ($IC_{50}$: 1.5 nM, combined $IC_{50}$: 1.4 nM) without any signs of cytotoxicity up to 20 μM. Lonafarnib blocked secretion of progeny virus to background levels with an $IC_{50}$ of 36 nM. Cytotoxicity was observed at concentrations >6 μM. Remarkably, in the primary infection, we found a dose-dependent increase in the in-cell ELISA HDAg signal, indicating an intracellular accumulation of HDAg in Lonafarnib-treated cells. The immune modulator IFN-α inhibited primary infection to about 40% of an uncompeted infection ($IC_{50}$: 17 IU/ml). Secondary infection was similarly decreased but not abrogated ($IC_{50}$: 38 IU/ml, combined $IC_{50}$: 28 IU/ml).

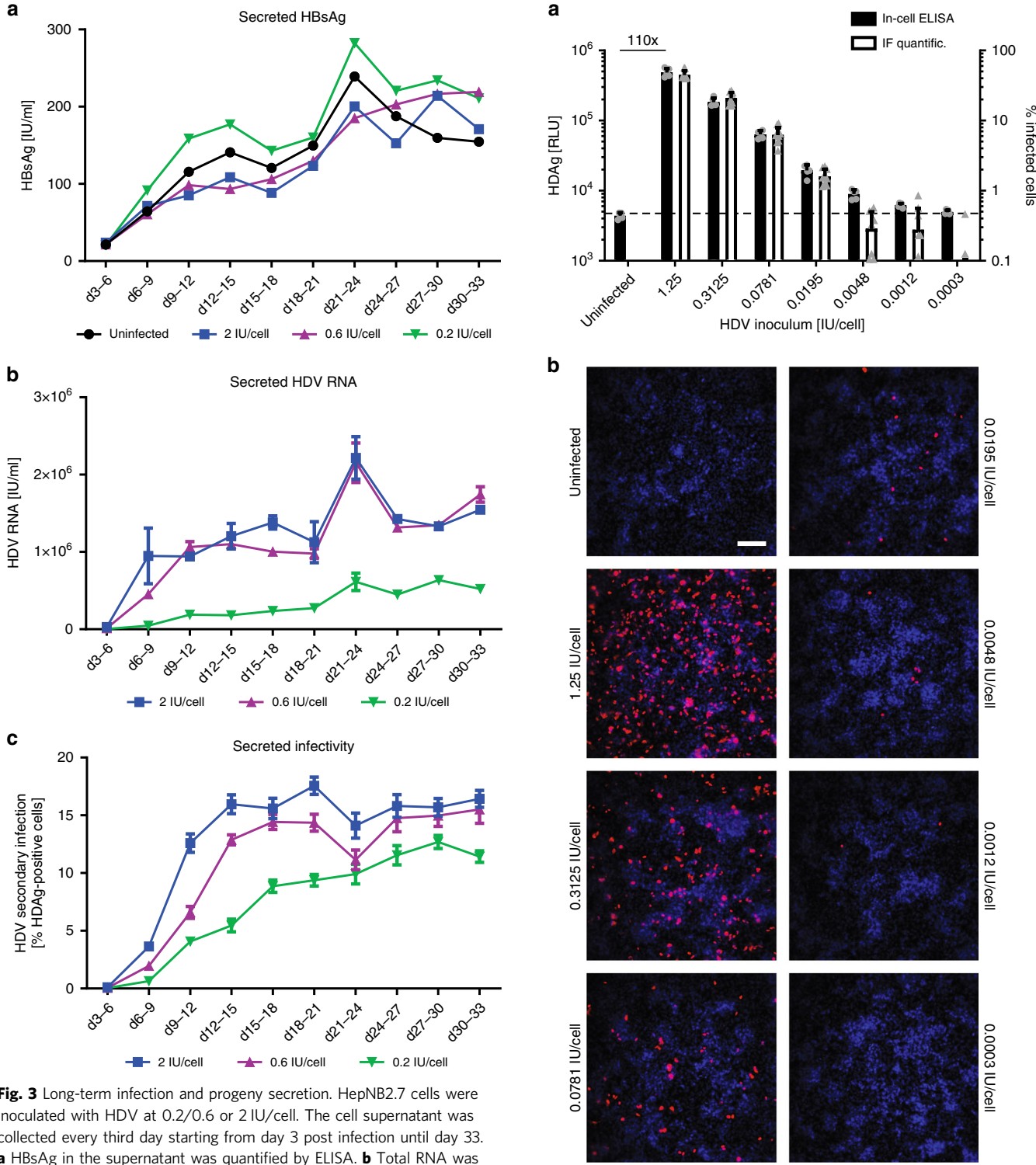

**Fig. 3** Long-term infection and progeny secretion. HepNB2.7 cells were inoculated with HDV at 0.2/0.6 or 2 IU/cell. The cell supernatant was collected every third day starting from day 3 post infection until day 33. **a** HBsAg in the supernatant was quantified by ELISA. **b** Total RNA was extracted from the cell supernatant and HDV RNA was quantified by RT-qPCR. **c** The supernatant of the respective time points was used as an inoculum for a secondary infection on HuH7-NTCP cells. HDAg-positive cells of the secondary infection were quantified by automated image analysis

**Fig. 4** Fast and reliable quantification of HDV infection by in-cell ELISA. HepNB2.7 cells seeded in a 96-well plate were inoculated with fourfold dilutions of HDV starting from 1.25 IU/cell. Cells were fixed at day 8 post infection and (**a**, black bars) in-cell ELISA was performed or (**b**, scale bar: 100 μm) immunofluorescence staining of HDAg was performed and positive cells were quantified by automated image analysis (**a**, white bars). The hatched line marks the in-cell ELISA signal level of the uninfected control to serve as the baseline

**Prenylation inhibition leads to intracellular accumulation of L-HDAg and viral genomes**. To elucidate the increase in HDAg as measured by in-cell ELISA for Lonafarnib-treated cells during primary infection, we treated HepNB2.7 cells with the 50-fold

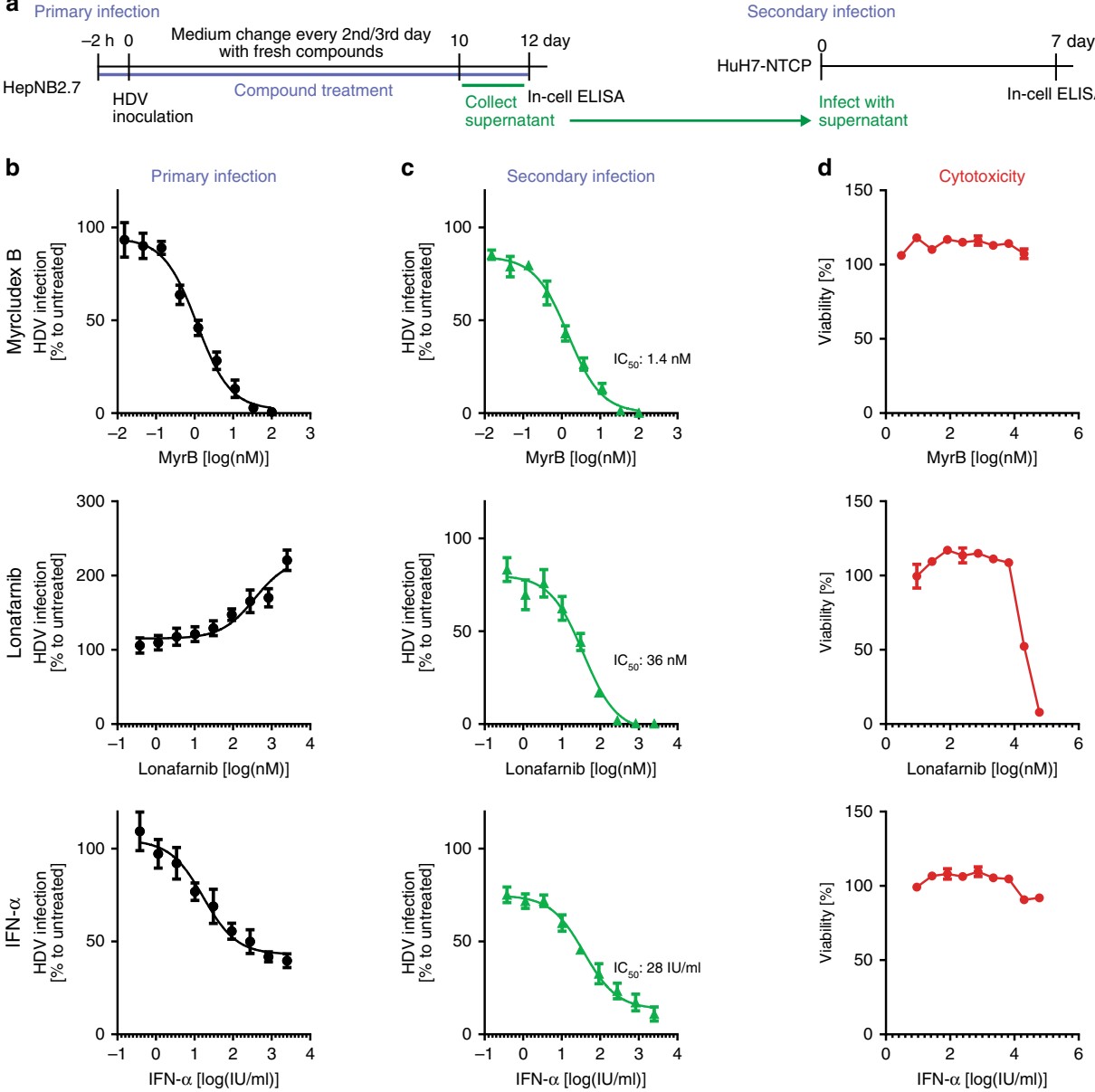

**Fig. 5** Antiviral drug evaluation on primary and secondary infection. **a** Schematic representation of the experimental layout: HepNB2.7 cells seeded in 96-well plates were pre-treated with the compounds for 2 h, and then inoculated with 0.3 IU/cell HDV in the presence of the compounds. The medium was replaced with fresh compounds every 2nd/3rd day. The supernatant from day 10 to 12 post infection was collected and one-fifth of the supernatant was used as an inoculum for secondary infection of HuH7-NTCP cells. **b** At day 12 post primary infection, cells were fixed and HDAg was quantified by in-cell ELISA. **c** HDAg of the secondary infection was quantified by in-cell ELISA at day 7 post secondary infection. All in-cell ELISA data from primary and secondary infection represent the mean of three independent infection experiments. **d** Cytotoxicity of the compounds was assessed on HepNB2.7 cells using WST-1 assay

$IC_{50}$ concentration of the respective drugs during HDV infection and quantified secreted progeny virus in a secondary infection (Fig. 6a). Consistent with the results in Fig. 5, Lonafarnib treatment led to an abrogation of progeny virus secretion. In comparison, IFN-α partially inhibited primary infection and in addition affected progeny release. However, both effects were weak (even 1400 IU/ml did not result in a complete block). Western blot analysis of the infected cells revealed an eightfold increase of intracellular L-HDAg in Lonafarnib-treated cells compared with the untreated controls, while the levels of S-HDAg only marginally (1.6-fold) increased (Fig. 6b). This indicates the accumulation of non-prenylated L-HDAg that cannot promote the envelopment of particles anymore. To gain more insight into

this accumulation, we generated an L-HDAg-specific polyclonal antibody by immunizing a rabbit with a peptide consisting of the C-terminal extension of L-HDAg. The antibody was validated in transfected cells expressing S-HDAg and L-HDAg, where it showed selective specificity for the L-antigen (Supplementary Fig. 6). The use of this antibody for immunofluorescence analysis verified the strong and selective increase of intracellular L-HDAg in Lonafarnib-treated cells (Fig. 6c). The staining pattern preferentially localized to the cytoplasm when compared with untreated cells. Finally, Lonafarnib treatment led to a twofold increase in the total intracellular HDV genomes (Fig. 6d). We and others have recently shown that HDV elicits strong innate immune responses in immunocompetent liver cells[12–14]. We

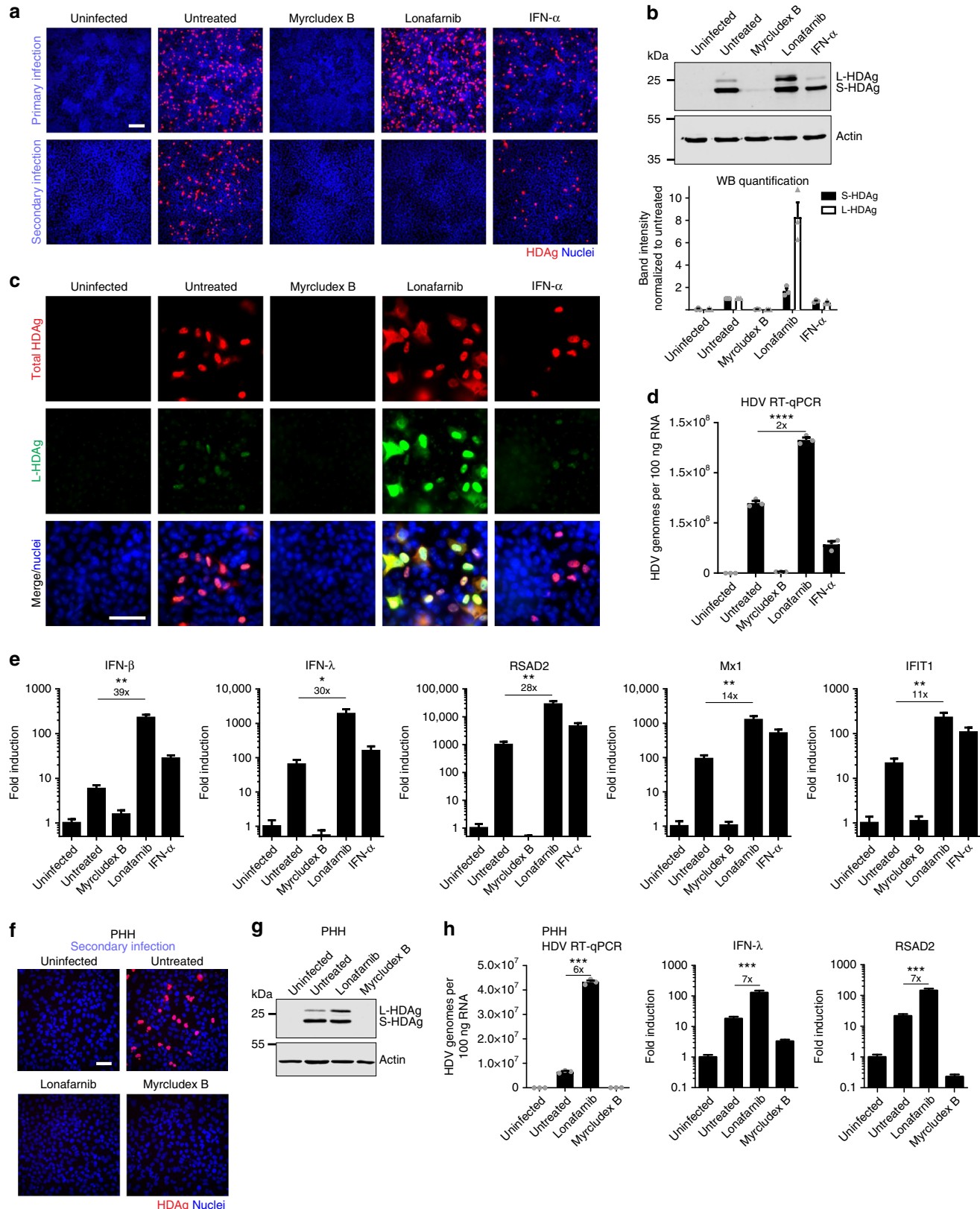

therefore investigated a possible induction of interferons (IFN-β, IFN-λ) and interferon-stimulated genes (ISGs: RSAD2, Mx1, and IFIT1) by RT-qPCR. As shown in Fig. 6e, HDV infection of HepNB2.7 cells led to an induction of IFN-β, IFN-λ, and ISGs, which is consistent with previous data and underlines the innate immune competence of the HepG2-cell line[14]. As expected, MyrB treatment blocked infection, and therefore no induction of IFN and ISGs was observed. Remarkably, treatment with Lonafarnib following infection increased the induction of both interferons and ISGs by 11- to 39-fold, which is probably due to the

**Fig. 6** Inhibition of prenylation leads to an enhanced innate immune activation. HepNB2.7 cells seeded in 24-well plates were pre-treated with 50× IC$_{50}$ concentration of the compounds for 2 h, and then inoculated with 0.5 IU/cell HDV in the presence of the compounds. The medium was replaced with fresh compounds every 2nd/3rd day. The supernatant from day 9 to 12 post infection was collected and one-fifth of the supernatant was used as an inoculum for secondary infection of HuH7-NTCP cells. **a** Cells of primary (top) and secondary (bottom) infection were fixed, and HDAg was immunostained (scale bar: 100 μm). **b** HDAg-specific western blot of the cells after primary infection treated with the indicated compounds (top). Western blot quantification of the two HDAg isoforms (bottom). Values represent the mean of three individual blots. **c** HepNB2.7 cells were fixed after primary infection and co-immunostained with a human serum for the total HDAg (red) and with a rabbit serum specifically for L-HDAg (green) (scale bar: 50 μm). **d, e** The total RNA was extracted from HepNB2.7 cells after primary infection, reverse transcribed, and the total HDV genomes, interferons, and ISGs were quantified by qPCR ($n = 3$ replicates). **f, g, h** Primary human hepatocytes (PHH) were co-infected with 0.5 IU/cell HDV and 150 ge/cell HBV and treated with 25× IC$_{50}$ concentration of the compounds throughout the infection. The medium was replaced with fresh compounds every 2nd/3rd day. The supernatant from day 10 to 12 post infection was collected and one-fifth of the supernatant was used as an inoculum for secondary infection of HuH7-NTCP cells (**f**, scale bar: 50 μm). The PHH of the primary infection at d12 were lysed and S- and L-HDAg were analysed by western blot (**g**). The total RNA was extracted from the PHH at d12, and the total HDV genomes, IFN-lambda, and RSAD2 copies were determined by RT-qPCR ($n = 3$ replicates) (**h**). Unpaired two-tailed Student's $t$ test was used for statistical analyses (*$P < 0.05$, **$P < 0.01$, ***$P < 0.001$, ****$P < 0.0001$, n.s. not significant, $P > 0.05$)

accumulation of intracellular HDV RNA. To verify these results under authentic conditions, we co-infected PHH with HDV and HBV in the presence of Lonafarnib and MyrB (25-fold IC$_{50}$), collected the supernatant between day 10 and 12 post infection, and analyzed the infectivity of progeny HDV (Fig. 6f). Western blot analysis of HDAg in the infected PHH again showed an increase in L-HDAg (Fig. 6g). Quantitative PCR on intracellular HDV replicative intermediates and the two markers for IFN induction (IFN-λ and RSAD2) again showed a profound increase in intracellular HDV replication and consequently increased the induction of IFN responses (Fig. 6h).

Taken together, the HepNB2.7 cell line supports the complete HDV replication cycle from viral entry to progeny virus secretion. In combination with an in-cell ELISA infection readout, we were able to verify the expected modes of action of the two most advanced HDV antiviral drugs, as well as the only current treatment option IFN-α. We determined the IC$_{50}$ of the prenylation inhibitor Lonafarnib and additionally found that blocking farnesylation of L-HDAg leads to an intracellular accumulation of HDAg and HDV RNA that is accompanied by an increase in innate immune induction.

## Discussion

As current cell culture models for HDV infection only support entry and intracellular replication[8,21] but not assembly and release of the virus, we established a stable cell line that supports the full viral life cycle by overexpressing both the receptor NTCP as well as the HBV surface proteins in HepG2 cells. Initially, we co-expressed NTCP and the HBV envelope proteins in HuH7-NTCP cells, since these have a higher susceptibility for HDV than HepG2-NTCP cells[21] (which may be associated with their innate immune incompetence). However, upon HDV infection of these cells, progeny virus secretion was unstable and low. We therefore established the system on a HepG2-cell backbone.

One astonishing observation was that stable co-expression of NTCP and the HBV envelope proteins within one cell was possible and led to two separated and functional proteins. As the preS1 region of the L-protein irreversibly binds to NTCP with a high affinity[28], co-expression of this ligand together with its receptor was expected to result in formation of an inactive intracellular complex, which is subsequently degraded. Such an observation has been described in DHBV-infected hepatocytes and leads to the downmodulation of the receptor (carboxypeptidase D)[29]. It is also well known as a mechanism of superinfection–exclusion in other enveloped viruses like HIV[30]. However, this was not observed for the HBV envelope proteins and the L-protein-specific receptor NTCP. This is consistent with the ability of HDV to superinfect HBV-infected hepatocytes. It

indicates that the L-protein, when co-expressed with M- and S-proteins under an authentic promotor control, is protected from interacting with NTCP during protein synthesis and transport. Alternatively binding competence may be inactivated before both proteins reach their final localization, e.g. the binding-competent N-terminal part of preS1 might be hidden in the membrane, until the assembled virion is secreted from the cell. The fact that SVPs are continuously secreted without inactivating NTCP argues for this shielding effect[31].

The HBV surface protein overexpressing cells were restricted for HBV but not HDV infection, indicating that the HBV surface proteins mediate a selective superinfection–exclusion for HBV independent from receptor downmodulation. It has been shown for DHBV that the DHBV L-protein mediates superinfection–exclusion[32] by different pathways, including one that is independent of CPD degradation, and there is evidence that the same mechanism is present in HBV infection (Yi Ni, unpublished results). The fact that HDV superinfection occurs indicates that the mechanism must target a step after viral fusion, e.g. transport of the capsid/ribonucleoprotein to the nucleus.

As HepNB2.7 cells support viral entry and assembly and release, they should also support viral spread. We tested this hypothesis over 22 days following a low-level infection. Within this period, at least two rounds of infection should occur, which would result in amplification of infected cells. However, the observed spread was minimal (1.2-fold) and could not be enhanced by PEG, pointing toward a restriction of viral spread in this system. It remains unclear if the limited spread is due to a constant activation of the innate immune system by HDV infection, blocking second-round infection of neighboring cells or if there are other cell-specific reasons, e.g. steric hindrances in HepG2 cells, which grow in large clusters. The latter hypothesis is supported by the observation that HBV spread in HepG2-NTCP is limited likewise.

In vitro analyses of the two investigational compounds Lonafarnib and Myrcludex B verified the mode of actions of these drugs in HepNB2.7 cells. Moreover, they allowed separating antiviral effects on early (entry and establishment of replication) and late infection events (replication and virus release). For IFN-α, the off-label used drug, we could show weak direct effects on both stages of infection. This indicates that, besides modulating the adaptive immune system, IFN-α also acts via multiple, yet unknown, direct pathways. However, even at very high concentrations, IFN-α is not able to abrogate HDV replication, which is consistent with previous in vitro findings[14] and the low rate of patients' responses treated with IFN.

Applying the HepNB2.7 cell line, we could assign the IC$_{50}$ for Lonafarnib (36 nM) and demonstrate the high efficiency of the drug to block cellular farnesyl transferase and the absolute

requirement of prenylation for HDV assembly. However, we also demonstrated that inhibition of L-HDAg prenylation does not repress HDV replication in hepatocytes. In contrast, we observed a profound accumulation of L-HDAg (eightfold) and HDV RNA replicative intermediates (twofold) within the cell. Our observation that similar effects were also observed in HBV/HDV co-infected PHH (Fig. 6f–h) strongly argues for accumulation of HDV replicative intermediates and an enhanced endogenous IFN response in hepatocytes of infected patients under Lonafarnib treatment. The results are in line with a previous report by Sato and colleagues, who used a different inhibitor of the farnesyl transferase (FTI-277) in a transfection-based model and also observed an increase of intracellular L-HDAg[33]. In general, these observations are consistent with a strong decline of HDV serum RNA levels in the Lonafarnib clinical trials[16]; however, it raises the question whether HDV RNA can be cleared from the liver if enhancement of intrahepatic replication is induced by the drug. One possibility could be related to an enhanced turnover of Lonafarnib-treated cells (e.g. by the immune system or HDV-mediated cytotoxicity). However, this has so far not been demonstrated and can be counteracted by the ability of the viral RNA to spread through cell division[34].

For Myrcludex B, we could verify the ability of the drug to completely and efficiently (IC$_{50}$: 1.4 nM) block the entry of HDV particles via the NTCP pathway. Consequently, second-round infections were also blocked with the same efficiency. As expected, but not verified so far, Myrcludex B had no influence on HDV replication, once the infection has been established. This demonstrates the ability of the Myrcludex to block intrahepatic spread of the virus via the extracellular route (while intracellular spread via cell division may still occur). The profound effects of Myrcludex B in clinical trials[35] indicate that extracellular amplification of viral RNA contributes significantly to viral persistence. Thus, synergistic effects may be expected by additionally blocking cell-to-cell spread.

With the HepNB2.7 cell line, we have established a powerful tool for high-throughput antiviral compound screenings but also for screenings to identify novel host factors (e.g. RNAi or CRISPR/Cas9 knockout screens). It can also be applied to deepen our understanding of the viral replication cycle, e.g. by deciphering the consequences of innate immune activation. HepNB2.7 cells allow studying replication fitness and response to antiviral drugs of different HDV genotypes and/or clinical isolates. Conceptually, the cell line can even be used to study and compare the surface proteins of different HBV genotypes, if respective cell lines with subgenomic constructs covering the same region of the respective HBV subtypes are generated. Finally, the cells allow visualization of viral replication and assembly, e.g. by FISH for genomic or antigenomic viral RNA or by fluorescent labeling of the HDAg using amber technology and click chemistry as described for the HBV capsid[36].

## Methods

**Cloning and cell line development**. The HB2.7 subgenomic fragment comprising the L-/M-/S-HBsAg and HBx ORFs of HBV genotype D was PCR-amplified from the plasmid pT7HB2.7[23] using the primers listed in Supplementary Table 1 and inserted into the lentiviral production plasmid pLX304[37], which had been digested with SmaI and NheI to remove the CMV promoter. Lentiviruses were produced and concentrated as previously described[21], and used to transduce HepG2-NTCP cells[21] (the parental HepG2 cells have been provided by Ralf Bartenschlager). A total of 15 single-cell clones were selected and characterized, regarding their ability to transport taurocholic acid, to secrete HBsAg, and to secrete infectious progeny HDV virions after infection. The best single-cell clone (#24) was selected and used for all subsequent experiments.

**Virus production and infection**. The HDV virus stock was produced by co-transfection of HuH7 cells with the plasmids pSVLD3 (HDV genotype 1, kindly provided by John Taylor[3]) and pT7HB2.7 (HBV genotype D envelope proteins, kindly provided by Camille Sureau[23]). The cell supernatant was harvested from day

8 to 14 post transfection; the virus was purified by heparin affinity chromatography, aliquoted, and stored at −80 °C until use. The viral titer was quantified by RT-qPCR using the World Health Organization HDV standard (Paul-Ehrlich-Institut, Langen, Germany). HBV was produced in HepAD38 cells[24]. Two weeks after removal of tetracycline from the culture medium, virus-containing cell supernatants were harvested, and virions were purified and concentrated by heparin affinity chromatography, aliquoted, and stored at −80 °C until use. For infection, cells were seeded in 96- or 24-well plates and cultured for 1–2 days, until they reached 80–100% confluence. Cells were inoculated with 0.3–0.5 IU/cell HDV (if not noted otherwise) in a medium containing 2% dimethyl sulfoxide (DMSO) and 4% polyethylene glycol (PEG) 8000 (Sigma). After 16 h of inoculation, the cells were washed twice with PBS, and fresh medium containing 2% DMSO was provided and replaced every second or third day until the end of the experiment. For drug treatment experiments, cells were pre-incubated at the indicated concentrations of Myrcludex B (Bachem), Lonafarnib (MedChem Express), or IFN-α (IFN-alpha2A, PBL) for 2 h before inoculation, as well as during and post inoculation. New drugs were added throughout every medium exchange, until the end of the experiment. For secondary infections, HuH7-NTCP cells were inoculated with one-tenth or one-fifth of the cell culture supernatant from the primary infection (if not noted otherwise) in a medium containing 4% PEG and 2% DMSO for 16 h. Cells were fixed and analyzed at day 6 or 7 post infection.

**In-cell ELISA**. Cells were cultured in white, non-transparent 96-well plates to reduce luminescence spillover. At the end of the infection experiment, cells were fixed in 4% PFA for 30 min, followed by 30 min of incubation in permeabilization buffer (PBS, 0.25% Triton X-100). After 30 min of incubation in blocking buffer (PBS, 0.05% Tween-20, 3% BSA), cells were incubated for 2 h with a patient serum containing anti-HDAg antibodies (VUDA) diluted in blocking buffer. After washing, endogenous peroxidases were blocked by 10 min of treatment with 3% (HuH7) or 1% (HepG2) hydrogen peroxide solution. Cells were incubated with a secondary goat–anti-human–HRP (Jackson Immuno Research) antibody diluted in blocking buffer for 1 h. After extensive washing, a chemiluminescence substrate (Advansta ELISABright) was added to the wells, and luminescence was measured on a plate reader.

**L-HDAg-specific antibody generation and immunofluorescence**. For the generation of polyclonal antibodies specific for L-HDAg, we synthesized a peptide corresponding to the C-terminal extension of L-HDAg (sequence: QGFPWDILFPADPPFSPQSCRPQ). The peptide was coupled to KLH and used to immunize a rabbit according to standard protocols (Davids Biotechnologie, Regensburg, Germany). For immunofluorescent staining, cells were fixed with 4% PFA for 30 min, followed by permeabilization in PBS/0.25% Triton X-100 for 30 min. Cells were incubated with a patient serum containing anti-HDAg antibodies (VUDA, 1:3000 dilution) for staining of the total HDAg or with the rabbit serum for staining of L-HDAg (1:5000 dilution) diluted in PBS/5% skim milk powder overnight (the VUDA serum was kindly provided by Heiner Wedemeyer, Hannover. The serum sample has been obtained with informed consent of the patient). After washing, secondary antibodies goat–anti-human-555 or goat–anti-rabbit-488 (Invitrogen) were added for 1 h, and the cells were imaged on an inverted epifluorescence microscope.

**RT-qPCR**. RNA was extracted from the cells using the Nucleospin RNA kit (Macherey-Nagel) and from cell supernatants using the QIAamp viral RNA mini kit (Qiagen) according to the manufacturer's instructions. The HDV RNA secondary structure was melted by incubation of the RNA at 95 °C for 5 min, followed by immediate cooling down to −80 °C. Quantification of HDV RNA in the cell supernatant was performed using the Luna universal probe One-Step RT-qPCR kit (New England Biolabs) and the primers and probe described by Ferns et al.[38]. qPCR values were normalized to the World Health Organization HDV standard (Paul-Ehrlich-Institut, Langen, Germany). For all other quantification, RNA was reverse transcribed using the High-capacity cDNA reverse transcription kit (Applied Biosystems) according to the manufacturer's instructions. Intracellular HDV RNA was quantified using the PerfeCTa qPCR ToughMix (Quanta Biosciences) and the primers and probe as described above. Induction of interferons and ISGs was quantified using the iTaq Universal SYBR Green Mix (Bio-rad) and the primers listed in Supplementary Table 1. All qPCRs were run on a CFX96 thermocycler (Bio-rad). Induction levels of interferons and ISGs were normalized to GAPDH.

**WST-1 cell viability assay**. HepNB2.7 cells were incubated with the compounds at the indicated concentrations for 4 days. WST-1 solution (Roche) was diluted 1:20 in a medium and incubated with the cells for 30 min. Absorption was read at 450 nm using a multiwell-plate reader and normalized to an untreated and total lysis control.

**Western blot**. Cell lysates were applied to SDS-PAGE on 12% or 15% SDS gels. For NTCP-specific WBs, cell lysates were deglycosylated before SDS-PAGE with PNGase F (New England Biolabs) according to the manufacturer's protocol. Proteins were transferred onto nitrocellulose membranes by semidry transfer and incubated with primary antibodies (HDAg: serum of a chronically infected patient [VUDA, 1:3000 dilution]; HBsAg: human–anti-HBsAg [Humabs Biomed, kind gift from Davide Corti, 1:3000 dilution]; NTCP: rabbit–anti-NTCP [Sigma, HPA042727, 1:1000 dilution]; actin: mouse–anti-actin [Sigma, A1978, 1:5000

dilution]) diluted in 5% skim milk/TBST overnight at 4 °C. Membranes were washed with TBST and incubated with fluorescently labeled secondary antibodies (LI-COR Biosciences) for 1 h at RT, washed again, and imaged on a LI-COR Odyssey imaging system.

**Taurocholate uptake assay.** Cells were seeded in 24-well plates 1 day prior to the experiment. The culture medium was removed, and cells were incubated in standard DMEM medium in the presence or absence of 2 μM MyrB for 15 min at 37 °C. A mix of 50 μM unlabeled TC (Sigma) and 5 nM [3H]-TC with a specific activity of 10 Ci/mmol (Hartmann Analytic) was added and co-incubated for another 15 min at 37 °C. Cells were incubated on ice, washed three times with ice-cold PBS to stop uptake, and lysed with PBS/0.05% sodium dodecyl sulfate/0.25 M NaOH. The lysate was mixed with Ultima Gold liquid scintillation solution (Perkin Elmer), vortexed extensively, and [3H] scintillation counts were determined using the LS 6000 liquid scintillation counter (Beckman Coulter).

**HBsAg/HBeAg quantification.** Secreted hepatitis B e antigen (HBeAg) was quantified in the cell supernatant by the ADVIA Centaur XPTM automated chemiluminescence system (Siemens), and secreted hepatitis B surface antigen (HBsAg) was determined by enzyme-linked immunosorbent assay (Architect, Abbott).

**Statistical analysis**. If not stated otherwise, experiments were repeated at least three times, and one representative dataset is shown. All error bars throughout the study represent the standard error of the mean (SEM). Statistical significance was determined using unpaired two-tailed Student's t test for single comparisons in the software GraphPad Prism 6 (*$P < 0.05$, **$P < 0.01$, ***$P < 0.001$, ****$P < 0.0001$, n.s. not significant, $P > 0.05$).

**Reporting summary**. Further information on research design is available in the Nature Research Reporting Summary linked to this article.

## Data availability

Uncropped western blot scans are compiled in Supplementary Fig. 7. The source data underlying Figs. 1c, d, 2c, d, 3a, c, 4a, 5b–d, 6b, d, e, and h and Supplementary Figs. 1, 2b, 3b, 4b, and 5a are provided as a Source Data file. All further relevant source data that support the findings of this study are available from the corresponding author upon reasonable request.

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

## Acknowledgements

This work received funding by the German Centre for Infection Research (DZIF), TTU Hepatitis, Project 5.807 and 5.704, the Heidelberg Cellnetworks Cluster of Excellence, and the Deutsche Forschungsgemeinschaft (DFG) TRR179 (TP 15 and TP 18). We thank Christa Kuhn for clonal selection of the cell line, Christina Kaufman for peptide synthesis, Camille Sureau for providing the HBV envelope protein plasmid, and Ralf Bartenschlager for his continuous support.

## Author contributions

F.A.L. initiated the study, planned, conducted, analyzed experiments, and wrote the paper. F.S., L.R., L.N. and C.L. conducted and analyzed experiments, Y.N. and Z.Z. provided essential conceptual input and materials, S.U. supervised the study, analyzed data, and wrote the paper with contributions and approval from all authors.

## Additional information

**Competing interests:** S.U. is a co-applicant and co-inventor on patents protecting Myrcludex B as an HBV/HDV entry inhibitor. The remaining authors declare no competing interests.

