## [Peer Review File · Nature Communications]

Reviewers' Comments:

Reviewer #1:

Remarks to the Author:

Of 257 million chronic HBV carriers 20 million are co-infected with HDV. Chronic hepatitis delta (CHD) can cause very severe liver disease and frequently leads to development of hepatocellular carcinoma. Currently, there are direct-acting antivirals that have been approved for the treatment of CHD. Progress in understanding the HDV virology and development of better treatments has been hampered by the scarcity of adequate experimental systems. Primary hepatocytes and few cell lines have been shown to support HBV and HDV infection but these platforms are hampered by poor permissivity and low throughput. The discovery of hNTCP as a bona-fide receptor for HBV and HDV has led to the creation of significantly improved in vitro models. Here, the authors describe the characterization of a new cell lines, termed HepNB2.7 which co-expresses NTCP and the small, medium and large forms of the HBV surface protein. Lempp et al demonstrate that such HepNB2.7 cells release infectious HDV following infection for at least 30 days. In the HepNB2.7 model HDV infection was associated with induction of IFN β , IFN λ and several interferon-stimulated genes (ISGs). The authors further describe the generation of a new, highly sensitive in-cell ELISA-based assay for the detection and quantification of HDV infected cells. This assay, which can be downscaled to 96 well plate format, will undoubtedly very useful for the field. Lempp et al. use these two technical innovations to re-establish that the entry inhibitor MyrB and the prenylation inhibitor Ionafarnib efficiently suppress HDV infection. MyrB - somewhat expectedly - also prevented viral spread which is consistent with clinical data. The authors confirmed that LNF blocked HDV egress which corroborates the decline in RNA levels in LNF-treated patients. Notably, LNF treatment did not completely abrogate HDV RNA replication in the HepNB2.7 model which has potentially some implications on whether LNF can actually be used to cure CHD.

Overall, the study is well designed and the data largely support the claims by the authors. While not entirely novel HepNB2.7 cells are certainly a convenient system to study HDV infection and spread in a hepatoma cell line. The authors provide proof-of-concept in sensible experiments.

Specific comments:

Abstract: It is an overstatement that current cell culture models do not support assembly and secretion of HDV. As the authors are certainly well aware there are numerous studies showing co-transfection of HDV and HBV encoding plasmids can lead to the secretion of infectious HDV.

Page 3, line 49: In vivo HDV is highly hepatotropic because of its dependency on HBV and the use of shared - largely liver-specific - receptor NTCP. However, as the authors are most certainly aware experimental HDV infection has been shown in hNTCP expressing HeLa, Vero and CHO cells, all of which are of non-hepatic origin.

A shortcoming of the HepNB2.7 system is that it does not seem to support efficient viral spread. How do the authors explain the limited spread phenotype? Does addition of PEG during the culture enhance viral spread as was suggested for HBV (see e.g. Michailidis et al. (2017))

Figure 1: How does the HBsAg secretion/expression compare to that in HBV infected primary hepatocytes?

Figure 3: How often have these experiments been repeated? Are the spikes in secreted HBsAg and HDV RNA at days 21-24 reproducible?

Figure 4: please define the hatched line in figure 4

Figure 6: limit of detection in the panels in figure 6d is missing

Style: avoid the use of "interestingly" throughout the document

Hepatitis delta virus and hepatitis B virus should be spelled all lower case.

Methods: Please provide details on the HBV virus production, WB, taurocholate uptake assay, HBsAg and HBeAg assays etc. The reader should not have to dig through other papers to get detailed protocols in order to reproduce any of the data.

Reviewer #2:

Remarks to the Author:

In this study, Lempp and colleagues developed a novel HepG2-NTCP derived cell line stably overexpressing the envelope proteins of HBV to recapitulate the full HDV life cycle. They first demonstrated that this cell line was able to produce infectious HDV particles following virus infection, and this secretion was still observed more than 30 days after viral inoculation. Aiming to develop a method for the screening of antivirals, they then set up an ELISA-based system to quantify HDAg production in HDV-infected cells. Using this system, they determined the IC₅₀ of the prenylation inhibitor Lonafarnib, and described an unexpected increase in HDAg levels in Lonafarnib-treated cells. Finally, using a specific anti-L-HDAg antibody, they showed that this increase was linked to an accumulation of L-HDAg and viral genomes in treated cells. The study presents an elegant system for the study of HDV full life cycle. The manuscript is well written and the data clearly presented. However, a better characterization of the model, validation experiments using alternative systems and an application of this model (e. g. a drug screen with discovery of a novel antiviral or target) would increase the significance and impact of the study.

Specific comments.

1. The authors present an innovative system for the study of the full life cycle, but one key aspect of the model needs to be clarified. What is the rationale for the use of such a high fragment of the HBV genome? In the current system, HBx is expressed and may interfere with several processes. How is HBx expressed in HepNB2.7 cells? In the same vein, the authors claim in the discussion that the presence of HBs proteins may be responsible for the inhibition of HBV infection. In this context, the authors should also discuss the putative interactions between HBx and the incoming HBV particle.
2. The authors did not observe any viral spread in their system. One explanation may be the absence of PEG in the culture medium, which is usually used to enhance HBV and HDV infection in NTCP-derived cell culture systems. The experiment presented in Supplementary Figure 1 should be reproduced with increasing concentrations of PEG in the culture medium. In the same vein but in absence of infection, as the authors described no interference between secreted HBs proteins and bile acid uptake, what would be the consequences of the addition of PEG on bile acid uptake in HepNB2.7 cells?
3. Figure 3. The authors claim that a maximum of infection is reached, given that the level of secreted HDV RNA is comparable between 2 IU/cell and 0.6 IU/cell. The authors should confirm this point using high MOIs (10; 100...). Moreover, the authors should put this statement in perspective with the relatively limited level of infection (15-30% of infected cells).

4. While the authors developed an elegant infection system for the set-up of innovative anti-HDV screens, they only applied it for the characterization of known HDV inhibitors Myrcludex B and Lonafarnib. To demonstrate the robustness and impact of their model, the authors should present a proof-of-concept pilot screen using small molecule library for the identification of novel inhibitors of HDV infection or re-infection.

Moreover, while their model is the starting point for the study of the full life cycle, including unexplored aspects of virus-host interactions, they used it to characterize HDAg accumulation following the inhibition of L-HDAg prenylation, which can be studied in regular NTCP-expressing cells. Did the authors try to inhibit the different cellular secretory pathways to determine how HDV particles are secreted from infected cells?

5. The presented results would benefit from being validated in physiological models, such as primary human hepatocytes, as a model of secondary infection. In the same vein, the observed effect of Lonafarnib on HDAg accumulation should be validated in PHH.

6. Contrary to what is claimed in the reporting summary, the number of independent experiments as well as the total number of replicates per experiment ("n") is not indicated in the manuscript. The authors should clarify that point. Moreover, why no statistical test was used in this study?

Reviewer #3:

Remarks to the Author:

NCOMMS-18-29805

Lempp et al describe the development of a HepG2 cell line stably expressing both NTCP, the receptor for hepatitis B and hepatitis delta viruses, and the hepatitis B virus envelope proteins. The authors show that this cell line can produce infectious HDV; it can also be infected by HDV produced from these cells or from other sources. These activities have been demonstrated individually for different cell lines, but this report is the first to show that the same cell line supports both processes. Unfortunately, for reasons that are not yet clear, the cells do not support spread of the virus through the culture to an appreciable extent. Intriguingly, the authors also demonstrate that these cells cannot be infected by HBV. This pattern – infectious for HDV but not HBV – appears to mirror what occurs during natural infection. This result is likely to trigger further studies into how the entry processes for these two viruses differ following attachment to the same cell surface receptor. The authors then presented an in-cell ELISA assay to quantify the number of cells infected and used this assay to demonstrate that these cells may be used for important practical purposes, such as quantitative assessment of antiviral activity. These are important technical advances that will be highly useful for discovery of new HDV inhibitors and identification of host processes involved in HDV infection. One snippet that might be overlooked in this wide-ranging manuscript is the substantial increase in innate responses to HDV that occurred following lonafarnib treatment; this increase is likely related to effects of lonafarnib on aspects of HDV replication. The observed effect of lonafarnib on innate responses is important because it could certainly add to its clinical efficacy. Overall, this is a thoroughly sound study that will be of significant interest to those in the HDV and HBV fields.

Major points to address:

None

Minor points:

1. Sato et al. (2004) previously showed that a farnesyl transferase inhibitor can increase the amount of L-HDAg produced. This work should be cited.

John Casey

Point by point response

Reviewer #1:

Of 257 million chronic HBV carriers 20 million are co-infected with HDV. Chronic hepatitis delta (CHD) can cause very severe liver disease and frequently leads to development of hepatocellular carcinoma. Currently, there are direct-acting antivirals that have been approved for the treatment of CHD. Progress in understanding the HDV virology and development of better treatments has been hampered by the scarcity of adequate experimental systems. Primary hepatocytes and few cell lines have been shown to support HBV and HDV infection but these platforms are hampered by poor permissivity and low throughput. The discovery of hNTCP as a bona-fide receptor for HBV and HDV has led to the creation of significantly improved in vitro models. Here, the authors describe the characterization of a new cell lines, termed HepNB2.7 which co-expresses NTCP and the small, medium and large forms of the HBV surface protein. Lempp et al demonstrate that such HepNB2.7 cells release infectious HDV

following infection for at least 30 days. In the HepNB2.7 model HDV infection was associated with induction of IFN β , IFN λ and several interferon-stimulated genes (ISGs). The authors further describe the generation of a new, highly sensitive in-cell ELISA-based assay for the detection and quantification of HDV infected cells. This assay, which can be downscaled to 96 well plate format, will undoubtedly very useful for the field. Lempp et al. use these two technical innovations to re-establish that the entry inhibitor MyrB and the prenylation inhibitor lonafarnib efficiently suppress HDV infection. MyrB - somewhat expectedly – also prevented viral spread which is consistent with clinical data. The authors confirmed that LNF blocked HDV egress which corroborates the decline in RNA levels in LNF-treated patients. Notably, LNF treatment did not completely abrogate HDV RNA replication in the HepNB2.7 model which has potentially some implications on whether LNF can actually be used to cure CHD. Overall, the study is well designed and the data largely support the claims by the authors. While not entirely novel HepNB2.7 cells are certainly a convenient system to study HDV infection and spread in a hepatoma cell line. The authors provide proof-of-concept in sensible experiments.

1.1) Abstract: It is an overstatement that current cell culture models do not support assembly and secretion of HDV. As the authors are certainly well aware there are numerous studies showing co-transfection of HDV and HBV encoding plasmids can lead to the secretion of infectious HDV.

We agree with the reviewer and changed this sentence in order to make clear that we are referring to infection- rather than transfection-based cell culture models.

1.2) Page 3, line 49: In vivo HDV is highly hepatotropic because of its dependency on HBV and the use of shared – largely liver-specific – receptor NTCP. However, as the author are most certainly aware experimental HDV infection has been shown in hNTCP expressing HeLa, Vero and CHO cells, all of which are of non-hepatic origin.

We agree with the reviewer and implemented an additional sentence in the introduction.

1.3) A shortcoming of the is that it does not seem to support efficiently viral spread. How do the authors explain the limited spread phenotype? Does addition of PEG during the culture enhance viral spread as was suggested for HBV (see e.g. Michailidis et al. (2017))

We performed a 22-days-cultivation of HepNB2.7 cells in the presence of 4% PEG with and without Myrcludex B. As shown in Suppl. Fig. 4, the number of HDVAg positive cells of the untreated group was only slightly higher than the Myrcludex B treated group indicating that addition of PEG during the culture does not significantly enhance the viral spread. Our previous work already demonstrated that addition of PEG in the inoculum medium only slightly increases the infection rate of HDV in HepG2-NTCP cells (in contrast to its strong enhancement of HBV infection). This is consistent with the additional results in the HepG2-derived HepNB2.7 cells.

One explanation of the limited spreading efficiency of HDV in the HepNB2.7 is a fast decline of the susceptibility after confluency of the cells. As shown in Suppl. Fig. 3, the susceptibility of HepNB2.7 to HDV decreased >80% at d11 post seeding in comparison to d1 post seeding. This is probably a major restriction factor for the low HDV spread considering the slow kinetics of progeny virus production. Another possible restriction affecting virus spread is the innate immune response induced by HDV replication in HepG2-derived cell lines. Our previous work showed that IFN treatment from day 1 post infection (p.i.) on can significantly decrease the infection rate in HepG2-NTCP cells although the effect is marginal if the treatment starts after the establishment of replication (from day 5 p.i. on). Thus, pre-existence

of innate immune responses can inhibit the establishment of productive infection in the HepNB2.7 system.

1.4) Figure 1: How does the HBsAg secretion/expression compare to that in HBV infected primary hepatocytes?

In this manuscript, using HepNB2.7 cells, we observed a time-dependent increase of secreted HBsAg between 20 – 200 IU/ml after seeding. This corresponds to the kinetics of HBsAg secretion in HepG2.2.15 cells and HepAd38 cells (both encode HBsAg in the integrate and express it under authentic promotor control). In this regard, the in vitro models correlate quite well. However, we previously described that the levels of secreted HBsAg after in vitro HBV infection of PHH are much higher than the levels obtained from infected HepG2-NTCP cells. (Ni et al., Gastroenterology 2014, Schulze et al., Hepatology, 2012; Nkongolo et al., J Hepatol, 2014; Lempp et al., Hepatology, 2017). We presently don't understand why that is but speculate that assembly of HBsAg in hepatoma cell lines is somehow restricted, and may lead to a "kinetic delay" regarding the particle formation.

1.5) Figure 3: How often have these experiments been repeated? Are the spikes in secreted HBsAg and HDV RNA at days 21-24 reproducible?

These experiments have been repeated 5 times, however, with modifications in inoculum size, total duration of experiments and readouts used. The spike at days 21-24 has not been observed in the other repetitions, therefore we did not specifically mention or discuss it in the manuscript.

1.6) Figure 4: please define the hatched line in figure 4

The hatched line marks the readout level of the uninfected control to serve as baseline for better comparison with the infected samples. We included the description in the figure legend.

1.7) Figure 6: limit of detection in the panels in figure 6d is missing

The detection limit for HDV qPCR is 100 copies per reaction. Since the number is too low to be clearly shown in the bar panels, we have indicated the detection limit in the figure legends of Fig.6.

1.8) Style: avoid the use of "interestingly" throughout the document

We deleted all "interestingly" from the manuscript.

1.9) Hepatitis delta virus and hepatitis B virus should be spelled all lower case.

We changed the manuscript accordingly.

1.10) Methods: Please provide details on the HBV virus production, WB, taurocholate uptake assay, HBsAg and HBeAg assays etc. The reader should not have to dig through other papers to get detailed protocols in order to reproduce any of the data.

We included the requested descriptions in the methods section.

Reviewer #2:

In this study, Lempp and colleagues developed a novel HepG2-NTCP derived cell line stably overexpressing the envelope proteins of HBV to recapitulate the full HDV life cycle. They first demonstrated that this cell line was able to produce infectious HDV particles following virus infection, and this secretion was still observed more than 30 days after viral inoculation. Aiming to develop a method for the screening of antivirals, they then set up an ELISA-based system to quantify HDAg production in HDV-infected cells. Using this system, they determined the IC₅₀ of the prenylation inhibitor Lonafarnib, and described an unexpected increase in HDAg levels in Lonafarnib-treated cells. Finally, using a specific anti-L-HDAg antibody, they showed that this increase was linked to an accumulation of L-HDAg and viral genomes in treated cells. The study presents an elegant system for the study of HDV full life cycle. The manuscript is well written and the data clearly presented. However, a better characterization of the model, validation experiments using alternative systems and an application of this model (e. g. a drug screen with discovery of a novel antiviral or target) would increase the significance and impact of the study.

2.1) The authors present an innovative system for the study of the full life cycle, but one key aspect of the model needs to be clarified. What is the rationale for the use of such a high fragment of the HBV genome? In the current system, HBx is expressed and may interfere with several processes. How is HBx expressed in HepNB2.7 cells? In the same vein, the authors claim in the discussion that the presence of HBs proteins may be responsible for the inhibition of HBV infection. In this context, the authors should also discuss the putative interactions between HBx and the incoming HBV particle.

We implemented such a large subgenomic HBV fragment (2.7 kB) for expression of the HBV envelope proteins in order to have the expression driven from the authentic HBV env promoters and to have an authentic mRNA that ends with the HBx polyA signal. Upon lentiviral integration of the subgenomic construct (there is no additional promoter in the

lentiviral plasmid) this construct and HBsAg expression from this construct should behave like that from integrated HBV DNA as it occurs in chronically infected patients. The reason for this design goes back to a large set of pre-experiments that were performed in order to define optimal conditions for HDV production by co-transfection. Here we side-by-side compared different plasmids for HBsAg expression, during virus production, e.g. we also had plasmids where only the HBsAg ORF driven by a SV40 promoter was present without additional 5' or 3' sequences. However, we found that the highest virus yield was achieved when the envelope proteins were expressed by the sub genomic HB2.7 construct rather than by artificial-promoter-driven plasmids. Therefore, we decided to use this construct not only for virus production but also for the generation of our HepNB2.7 cell line.

The reviewer is right that the HBx gene in this construct is not deleted or inactivated. Accordingly, HBx can principally be expressed in our cell line. Whether this is the case is unclear, since nobody succeeded so far to show HBx in cells that contain integrates (HepG2.2.15, Ad38 or others). However, we were aware of the reviewers concern beforehand and generated the same HB2.7 cell line encoding a stop-mutation in the HBx ORF. Comparing both constructs side-by-side, we did not observe any differences in HDV assembly and release (also no differenced ion HBsAg secretion. Thus, we can exclude a significant effect of HBx on HDV assembly & release. Since the focus of our paper is on HDV and not on possible effects of HBx in the context of an integrate we omitted to put the data of the respective X-minus cell line in. In order to provide this information for the reader we included a sentence in the text.

2.2) The authors did not observe any viral spread in their system. One explanation may be the absence of PEG in the culture medium, which is usually used to enhance HBV and HDV infection in NTCP-derived cell culture systems. The experiment presented in Supplementary Figure 1 should be reproduced with increasing concentrations of PEG in the culture medium. In the same vein but in absence of infection, as the authors described no interference between secreted HBs proteins and bile acid uptake, what would be the consequences of the addition of PEG on bile acid uptake in HepNB2.7 cells?

As already stated in response to reviewer 1 we performed an additional experiment implementing PEG and depict the new results in supplementary Fig.3. Addition of PEG (final concentration 4%), which is the optimal concentration to enhance HBV infection, did not significantly increase HDV spread (see argumentation for possible explanations in the response to reviewer 1). In addition, continuous inclusion of 4% PEG even decrease the

overall HDV infection rate possibly due to slightly toxic effects due to long term treatment. Regarding the bile acid uptake, we did not observe significant effect of PEG (final concentration 4%) on bile acid uptake in HepNB2.7 cells (see Suppl. Fig. 1).

2.3) Figure 3. The authors claim that a maximum of infection is reached, given that the level of secreted HDV RNA is comparable between 2 IU/cell and 0.6 IU/cell. The authors should confirm this point using high MOIs (10; 100...). Moreover, the authors should put this statement in perspective with the relatively limited level of infection (15-30% of infected cells).

We agree with the reviewer, that the limited level of infection, which cannot only be observed in HepNB2.7 but also in all other NTCP-overexpressing cell lines, presents an interesting observation that we cannot fully explain so far. We have already shown in a previous publication (Lempp et al., J Hepatol, 2016, Fig. 2) that the infection rate cannot be increased further by using inocula up to 10 IU/cell. In contrast, very high virus inocula even led to a lower level of HDAg-positive cells, which might be explainable by the induction of an increased antiviral innate immune response (see paper of Zhang et al., J Hepatol. 2018). As the reviewer pointed out, we did not mention this in the manuscript, therefore, we now added a respective statement in the results section.

2.4) While the authors developed an elegant infection system for the set-up of innovative anti-HDV screens, they only applied it for the characterization of known HDV inhibitors Myrcludex B and Lonafarnib. To demonstrate the robustness and impact of their model, the authors should present a proof-of-concept pilot screen using small molecule library for the identification of novel inhibitors of HDV infection or re-infection. Moreover, while their model is the starting point for the study of the full life cycle, including unexplored aspects of virus-host interactions, they used it to characterize HDAg accumulation following the inhibition of L-HDAg prenylation, which can be studied in regular NTCP-expressing cells. Did the authors try to inhibit the different cellular secretory pathways to determine how HDV particles are secreted from infected cells?

We fully agree with the reviewer regarding the suitability of the cells for medium and high throughput screening. In fact we have used the HepNB2.7 cells to screen a library of FDA-approved small molecule drugs and found one class of compounds, RAR-alpha agonists, that decreased secretion of HDV in the HepNB2.7 cells. After detailed mode-of-action studies, we found, however, that this class of compounds does not act directly on HDV but rather on the expression of HBsAg, both from integrates as well as from HBV cccDNA. The compounds

were further characterized for their HBV inhibitory potential in several different HBV cell culture systems and the paper is now under revision. We are happy to provide the reviewer with this manuscript upon request. However, due to the completely different focus of this drug screening paper we decided to not disclose the screening results in the manuscript presented here (but in the other paper). However, as reviewer 3 also pointed out, we believe that our study will trigger many subsequent studies that will use our cell line for both basic virology as well as further screening approaches.

2.5) The presented results would benefit from being validated in physiological models, such as primary human hepatocytes, as a model of secondary infection. In the same vein, the observed effect of Lonafarnib on HDAg accumulation should be validated in PHH.

We agree with the reviewer on the importance of such an experiment. To that aim we coinfecting PHH with HBV and HDV and analyzed the production of infectious progeny HDV as evidenced by the infectivity of the harvested supernatants. As expected from the results of the HepNB2.7 cells, the production of progeny viruses was almost completely blocked by the entry inhibitor Myrcludex B and the prenylation inhibitor Lonafarnib. In, Lonafarnib also increased the accumulation of the large HDAg, HDV RNA replication and the innate immune responses in PHH. These results are included in the revised Fig.6. We would like to stress that this experiment strongly supports the suitability of our system as a surrogate for such elaborated co-infection studies and confirms that Lonafarnib enriches replicative intermediates in hepatocytes.

2.6) Contrary to what is claimed in the reporting summary, the number of independent experiments as well as the total number of replicates per experiment (“n”) is not indicated in the manuscript. The authors should clarify that point. Moreover, why no statistical test was used in this study?

We thank the reviewer for pointing this out. We added a statistical section in the methods part, explaining experimental repetitions and furthermore, we added statistical testing to the data in figure 6.

Reviewer #3:

Lempp et al describe the development of a HepG2 cell line stably expressing both NTCP, the receptor for hepatitis B and hepatitis delta viruses, and the hepatitis B virus envelope proteins. The authors show that this cell line can produce infectious HDV; it can also be infected by

HDV produced from these cells or from other sources. These activities have been demonstrated individually for different cell lines, but this report is the first to show that the same cell line supports both processes. Unfortunately, for reasons that are not yet clear, the cells do not support spread of the virus through the culture to an appreciable extent. Intriguingly, the authors also demonstrate that these cells cannot be infected by HBV. This pattern – infectious for HDV but not HBV – appears to mirror what occurs during natural infection. This result is likely to trigger further studies into how the entry processes for these two viruses differ following attachment to the same cell surface receptor. The authors then presented an in-cell ELISA assay to quantify the number of cells infected and used this assay to demonstrate that these cells may be used for important practical purposes, such as quantitative assessment of antiviral activity. These are important technical advances that will be highly useful for discovery of new HDV inhibitors and identification of host processes involved in HDV infection. One snippet that might be overlooked in this wide-ranging manuscript is the substantial increase in innate responses to HDV that occurred following lonafarnib treatment; this increase is likely related to effects of lonafarnib on aspects of HDV replication. The observed effect of lonafarnib on innate responses is important because it could certainly add to its clinical efficacy. Overall, this is a thoroughly sound study that will be of significant interest to those in the HDV and HBV fields.

3.1) Sato et al. (2004) previously showed that a farnesyl transferase inhibitor can increase the amount of L-HDAg produced. This work should be cited.

We were not aware of this study and thank the reviewer for pointing it out. The citation has been included in the discussion section.

Reviewers' Comments:

Reviewer #1:

Remarks to the Author:

The authors have adequately address all the concerns I had raised during the previous round of review.

Reviewer #2:

Remarks to the Author:

In their revised version, the authors have addressed the key technical issues.